# An Overview of the Vision-Based Human Action Recognition Field

**Fernando Camarena *** , **Miguel Gonzalez-Mendoza *** , **Leonardo Chang** and **Ricardo Cuevas-Ascencio**

Tecnologico de Monterrey, School of Engineering and Science, Av. Eugenio Garza Sada 2501 Sur, Tecnológico, 64700 Monterrey, Mexico

*    Correspondence: fernando@camarenat.com (F.C.); mgonza@tec.mx (M.G.-M.)

**Abstract:** Artificial intelligence's rapid advancement has enabled various applications, including intelligent video surveillance systems, assisted living, and human–computer interaction. These applications often require one core task: video-based human action recognition. Research in human video-based human action recognition is vast and ongoing, making it difficult to assess the full scope of available methods and current trends. This survey concisely explores the vision-based human action recognition field and defines core concepts, including definitions and explanations of the common challenges and most used datasets. Additionally, we provide in an easy-to-understand manner the literature approaches and their evolution over time, emphasizing intuitive notions. Finally, we explore current research directions and potential future paths. The core goal of this work is to provide future works with a shared understanding of fundamental ideas and clear intuitions about current works and find new research opportunities.

**Keywords:** video-based human action recognition; action recognition; deep learning methods; handcrafted methods; human action; overview

## 1. Introduction

Artificial intelligence (AI) redefines our understanding of the world by enabling high-impact applications such as intelligent video surveillance systems [1], self-driving vehicles [2], and assisted living [3]. In addition, AI is revolutionizing areas such as education [4], healthcare [5], abnormal activity recognition [6], sports [7], entertainment [4,8], and human–computer interface systems [9]. These applications frequently rely upon the core task of video-based human action recognition, an active research field to extract meaningful information by detecting and recognizing what a subject is doing in a video [10–12]. Since its critical role in computer vision applications, the action recognition study can lead to innovative solutions that can benefit society in various ways. Nevertheless, it can take time to introduce oneself to the subject thoroughly.

On the one hand, current research points out numerous directions, including effectively combining multi-modal information [13,14], learning without annotated labels [15], training with reduced data points [15,16], and exploring novel architectures [17,18].

On the other hand, recent surveys shifted their focus towards comprehensively analyzing a particular contribution. For example, Ref. [8] categorized standard vision-based human action recognition datasets, whereas Ref. [19] analyzes the classification performance of standard action recognition algorithms. Ref. [20] was one of the first surveys to review deep learning algorithms, providing a comprehensive overview of the datasets employed. Ref. [21] offers a comprehensive taxonomy centered on deep learning methodologies, while Refs. [22,23] concentrates on its applicability. Ref. [24] explores human action recognition from visual and non-visual modalities. Ref. [25] provides proper taxonomy for action transformers according to their architecture, modality, and intended use. Ref. [26] evaluates existing solutions based on the computer vision challenge they solve. Ref. [22]

explores the action recognition field in conjunction with the related tasks of action detection and localization. Finally, Ref. [27] delved into future directions of the field.

Considering the vast expanse of knowledge and numerous potential directions within video-based human action recognition, introducing oneself to the subject requires significant time to develop a comprehensive understanding. As a result, this work strives to offer a comprehensive and intuitive overview of the vision-based human action recognition field:

- In Section 2, we start by defining core concepts, including definitions and explanations of the common challenges and most used datasets that may help future researchers have a shared understanding of the fundamental ideas.
- In Section 3, we break down the literature approaches and their evolution over time, emphasizing the intuitive notions that underpin the approaches' advancements. Therefore, future research may have a clear intuition of what researchers have proposed, and complex concepts make it more accessible to future works.
- In Section 4, we explore current research directions and potential future paths to help future works identify opportunities and boost the process to build further contributions. Finally, we discuss the conclusions in Section 5.

## 2. Understanding Video-Based Human Action Recognition

The aim of this section is threefold. First, Section 2.1 explains what this work understands as action. Second, we introduce the common challenges of video-based human action recognition in Section 2.2. Third, in Section 2.3, we introduce the commonly used datasets for action recognition. A summary can be found in Figure 1.

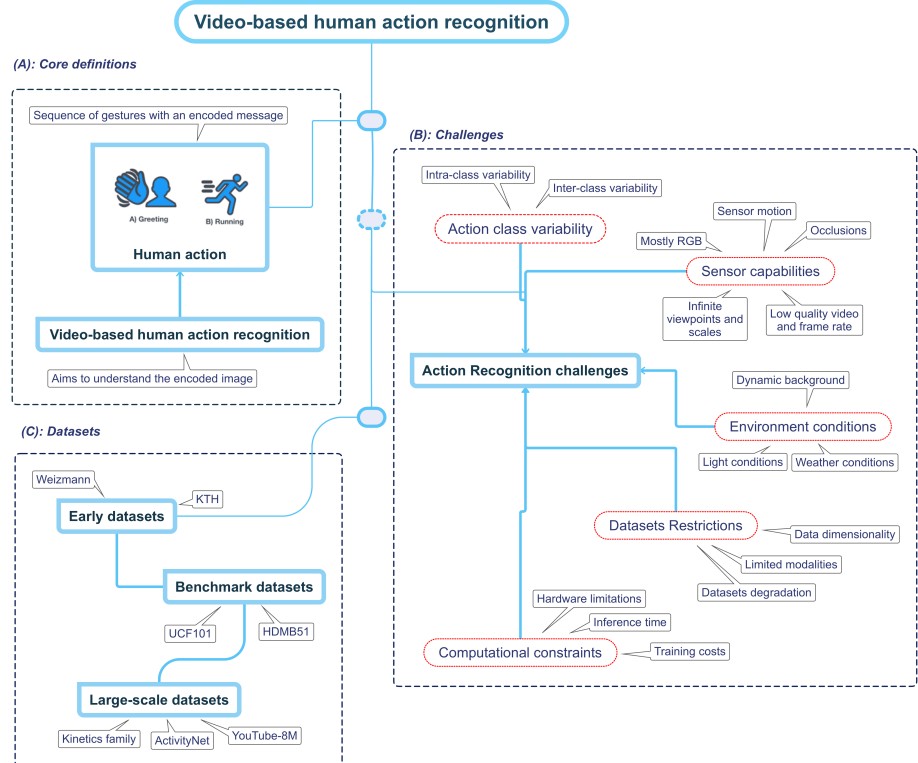

**Figure 1.** Video-based human action recognition overview. Part (**A**) represents human action; we instinctively associate a sequence of gestures with an action. For example, we might think of the typical hand wave when we think of the action greeting. On the contrary, imagining a person running will create a more dynamic scene with movement centered on the legs. Part (**B**) explains current challenges in the field, and Part (**C**) shows the relevant dataset used.

*2.1. What Is an Action?*

To understand the idea behind an action, picture the image of a person greeting another. Probably, the mental image constructed involves the well-known waving hand movement. Likewise, if we create a picture of a man running, we may build a more dynamic image by focusing on his legs, as depicted in Figure 1. We unconsciously associate a particular message with a sequence of movements, which we call "an action" [4,28]. In other words, human action is an observable entity that another entity, including a computer, can decode through different sensors. The human action recognition goal is to build approaches to understand the encoded message in the sequence of gestures.

Although it is a natural talent for a person to recognize what others do, it is not an easy assignment for a computer since it faces numerous challenges [20], explained in Section 2.2.

*2.2. Challenges Involved in Video-Based Human Action Recognition*

While humans have a natural ability to perceive and comprehend actions, computers face various difficulties when recognizing such human actions [26]. We categorize the challenges into five primary categories: action-class variability, sensor capabilities, environment conditions, dataset restrictions, and computational constraints. By understanding these challenges, we may build strategies to overcome them and, consequently, improve the model's performance.

2.2.1. Action Class Variability

Both strong intra- and inter-class variations of an action class represent a challenge for video-based human action recognition [26]. The intra-class variations refer to differences within a particular action class [29]. These variations stem from various factors, such as age, body proportions, execution rate, and anthropometric features of the subject [30]. For example, the running action significantly differs between an older and a younger individual. Additionally, we have repeated some of the actions so many times that we already perform them naturally and unconsciously, making it difficult even for the same person to act precisely the same way twice [26,30]. Finally, cultural contexts can impact how humans act, such as in the case of the greeting action class [31]. Due to the variability, developing a single model that accurately represents all instances of the same class is challenging [26]. Therefore, mitigating intra-class variation is a crucial research area in computer vision to represent all instances of the same class accurately.

Conversely, inter-class variation refers to the dissimilarities between distinct action classes [26], representing a significant challenge because some actions could share major feature vectors [27]. For example, while standing up and sitting down may be perceived as distinct actions, they share the same structure and semantics, making it challenging to differentiate one from another if the model approach does not consider their temporal structure [32]. A similar case is the walking and running actions, which, despite being different, can be seen as variations of the same underlying action. Therefore, to make computer vision applications more accurate and valuable, it is essential to make models that can handle inter-class variations.

2.2.2. Sensor Capabilities

In computer vision, a sensor detects and measures environmental properties such as light, temperature, pressure, and motion to convert them into electrical signals for computer processing [33]. Due to the capture of rich visual information, the RGB camera is the most common sensor used in video-based human action recognition, which senses the light intensity of three color channels (red, green, and blue) [4,33].

Using an RGB camera entails some challenges, including a reduced perspective due to the limited field of view [26], which may cause our target to be partially or not present in the camera field; a partial temporal view of the target subject is known as occlusion [4,26,34] and can be caused either by an object, another subject, the same subject or even the light conditions. Dealing with missing information is difficult because the occlusion may hide

the action's representative features [26]. For example, if a player's legs during a kick are not visible to the camera's field of view throughout a soccer match, it can be challenging to establish if they made contact with the ball.

Furthermore, there is no semantic of how to place the camera sensor, which implies that the target subject can appear in infinite perspectives and scales [35]. On the one hand, some perspectives may not help recognize an action [35,36]; for instance, when a person is reading a book, they will usually hold it in front of them; if the camera viewpoint is the subject's back, it will not perceive the book, and therefore, it will not be able to recognize the action.

On the other hand, our perception of speed is affected by the distance of the object from the camera [37]; even if two objects are moving at the same rate, but one of them is farther away from the camera, our brain will perceive that the farther objects are moving slower, an illusion known as "depth perception distortion" [37]. Earlier, we mentioned that running and walking actions differ in their temporal component, and this scaling effect can affect the accuracy recognition.

Another limitation is the low-video quality that some cameras feature [38], which can lead to a scenario where the target function represents only a few pixels that do not provide enough appearance information or the low camera frame rate does not capture the temporal nature of the action.

Although the camera has fixed placement, it does not imply that it is entirely static [39]; for instance, outdoor cameras are commonly affected by external factors that lead to image motions. Despite this, it may be imperceptible for a human. For a computer, it can be challenging because it may change the appearance features due to the lighting perception or misleading mix of the camera motion with the subject motion.

Another limitation is that the sensors extract only RGB images and, in some cases, audio [24]. Therefore, we are omitting complementary information that can boost the model's capabilities to represent an action class better [24].

### 2.2.3. Environment Conditions

Environmental conditions can significantly impact the classification accuracy of a model to recognize human actions by affecting the significance of the captured data [4,26]. To illustrate, poor weather conditions such as rain, fog, or snow reduce the target subject's visibility and affect the appearance features extracted. Likewise, in "real" conditions, the target subject will find itself in a scene with multiple objects and entities, which will cause a dynamic, unpredictable, and non-semantic background [26]; the delineation and comprehension of the objective and background can become increasingly complex and challenging when additional factors or variables are presented, which obscure the distinction between the foreground and background. Additionally, environmental conditions can generate image noise that limits representative visual features' extraction and complicates the subject track over time [40].

The environment light is also critical in identifying human actions [26], primarily if the model approach only relies on visual data for feature representation. Lighting conditions can cause subjects to be covered by shadows, resulting in occlusions or areas of high/low contrast, making taking clear, accurate, and visual-consistent pictures of the target subject complex. These circumstances may also result in images differing from those used during model training, confounding the recognition process even further.

### 2.2.4. Dataset Restrictions

The effectiveness of a machine learning model for recognizing human actions heavily depends on the dataset's quality used in its training phase [41]. The dataset's features, such as the number of samples, diversity, and complexity, are crucial in determining the model's performance. However, using a suitable dataset to boost the model's accuracy takes time and effort [42].

The first approach is constructing the dataset from scratch, ensuring the action samples fit the application's requirements. However, this process can be resource-intensive [42] because most effective machine learning models work under a supervised methodology, and consequently, a labeling process is required [43]. Data labeling [43] involves defining labeling guidelines, class categories, and storage pipelines to further annotate each action sample individually, either manually or by outsourcing to an annotation service to ensure consistent and high-quality data samples.

For some application domains, data acquisition can be challenging due to various factors [44], such as the unique nature of the application, concerns regarding data privacy, or ethical considerations surrounding the use of certain types of data [45]. Consequently, data acquisition can be scarce, insufficient, and unbalanced in the action classes, presenting significant obstacles to developing effective models or conducting meaningful analyses [44].

The second approach involves utilizing well-known datasets with a predefined evaluation protocol, enabling researchers to benchmark their methodology against state-of-the-art techniques. Nevertheless, there are some limitations, including the availability of labeled data; for example, the UCF101 [46] and HMDB51 [47] are one of the most used benchmark datasets [21]. Still, their data dimensionality is insufficient to boost the deep-learning model [48]. Furthermore, current datasets for action recognition face the challenge of adequately representing and labeling every variation of a target action [26], which is nearly impossible due to the immense variability in human movements and environmental factors. This limitation can impact the accuracy and generalizability of action recognition models if the dataset does not represent the same data distribution of the target application [26].

Another main problem with publicly available datasets is their degradation over time [26]; for example, a researcher that aims to use the kinetics dataset [48] must download each video sample from the Internet. However, some download links may no longer work, and specific videos may have been removed or blocked. As a result, accessing the same dataset used in prior research is impossible, leading to inconsistent results [26].

Most of the datasets provide the video along with a textual label tag [13]. Although this is enough to train a model to recognize human action, they have two main limitations. On the one hand, there is no clear intuition that text label tags are the optimal label space for human action recognition [49], particularly in cases where a more nuanced or fine-grained approach to labeling is required or in an application scenario where multi-modal information is available [13]. On the other hand, the exclusive use of RGB information in current datasets overlooks the potential benefits of other input sensors [24], such as depth or infrared sensors, which may provide more detailed and complementary representations of human actions in specific application scenarios.

### 2.2.5. Computational Constraints

The computational resources required to train and deploy a machine-learning model for video-based human action recognition can pose significant challenges for researchers [13].

Regarding model training, most approaches use a supervised methodology [42] whose performance depends on the data dimensionality, and hyperparameter tuning [50]. Consequently, they involve sophisticated architecture designs [51], leading to over-parameterized models requiring extensive computational resources [51]. The well-known model GPT-3 [52] comprises 175 billion parameters and is estimated to demand 3.14E23 FLOPS of computing power. If a V100 GPU were employed, it would require 355 GPU years to complete and cost roughly USD 4.6 million [52]. Additionally, sometimes researchers work with low-quality data; hence, they need to review and preprocess the dataset before the training process [53], which could be labor-intensive considering the data dimensionality required for model deep learning architectures.

Second, some application domains, such as video surveillance systems, require fast inference responses [27], which can be challenging because the model's complexity can exceed the processing capabilities of the underlying hardware [26,27]. To achieve fast

inference response, the model must analyze and classify video data in a time frame, almost in real-time [26,27].

Other application domains restrict to edge devices that prioritize small factors, portability, and convenience instead of processing power [54]. Some devices cannot perform high-end operations such as 3D convolutions [54].

*2.3. Datasets for Video-Based Human Action Recognition*

As the field of video-based human action recognition continues to grow, researchers increasingly rely on datasets to benchmark their proposed approaches [26], accelerate model development, and mitigate some of the challenges described in Section 2.2.

Finding a dataset that comprehensively covers all possible matches is nearly impossible. Consequently, the researchers must ensure the feature's dataset covers their application requirements.

Early works in RGB-based approaches used the KTH [55] and Weizmann [56] datasets, commonly known as constrained datasets [22]. Although the video clips provide valuable insights about the action samples, they may only partially represent the complexities and challenges of real-world scenarios since they were artificially recorded in controlled environments [22,32]. In the present state of video-based human action recognition, the KTH [55] and Weizmann [56] datasets no longer represent a challenge because current methods outperform them with nearly 100% of classification accuracy [22].

Conversely, due to the growth of video content on social media, including youtube and movie productions, researchers created datasets with a more comprehensive view of the complexity of human action in natural environments [26]. Two of the most common datasets are the UCF101 [46] and HMDB51 [47] datasets. On the one hand, the UCF101 [46] dataset gathered 13,320 video samples from the youtube dataset and divided it into 101 action categories that contain variations in camera motion, object appearance, and pose, object scale, viewpoint, cluttered background, illumination conditions to mitigate some of the challenges described in Section 2.2.

On the other hand, the HMDB51 [47] dataset consists of 6849 video samples extracted from multiple sources, including movies, Preminger archive, YouTube, and Google Videos, divided into 51 action classes. The HMDB51 [47] provides a comprehensive view of human action in a natural environment with variability in the illumination, subject appearance, and backgrounds.

To date, current approaches have achieved high-accuracy performance on the UCF101 [46] and HMDB51 [47]. Even though they still considered benchmark datasets in action recognition and related tasks [27], including self-supervised action recognition [43], zero-shot action recognition [57], and video generation [58], they have the central problem of data dimensionality since the number of video samples is not enough for deep learning requirements [48].

The Kinetics [48] dataset was introduced to address the limitations of existing action recognition benchmarks. The Kinetics [48] size was several orders of magnitude larger compared to UCF101 [46], and HMDB51 [47], including 400 action classes and 300 thousand video samples. Since its introduction, Kinetics has evolved into a family of datasets, including Kinetics-400 [48], Kinetics-600 [59], and Kinetics-700 [60], each containing at least 400, 600, and 700 video clips, respectively, for their corresponding number of action classes.

Continuing this trend, ActivityNet [61] introduced a large-scale video benchmark with 849 h of untrimmed videos of daily activities divided into 203 activity classes. Additionally, to the label tag, the activity net adds the temporal boundaries of the action sample in the video, which help other related tasks, including temporal action detection and action segmentation.

A prominent dataset is YouTube-8M [62], which contains 350,000 hours of videos with audio divided into 3862 classes. In addition to video-based human action recognition, the dataset can be used for understanding tasks, such as content-based video retrieval

and video summarization. A recent extension, Youtube-8M Segments [63], added 237,000 human-verified segment labels that make the dataset appropriate for temporal localization.

In addition to these RGB-based datasets, the NTU RGB+D [64], Kinectics-Skeleton [65] dataset, and J-HMDB [66] include depth and skeleton information, which can further aid in action recognition with additional information on the spatial and temporal features. On the other hand, MUGEN [67] is a novel dataset with 233,000 unique videos focused on multi-modal research, specifically to understand the relation between audio, video, and text. Finally, the something-something v2 dataset [49] contains 20,847 labeled videos of everyday actions that capture the granularity of video action.

## 3. The Evolution of Video-Based Human Action Recognition Approaches

This section provides an overview of the evolution of video-based human action recognition. We break down into two parts; first, in Section 3.1, we explain the first family of approaches known as handcrafted approaches. Second, in Section 3.2, we speak about the rise of deep learning approaches.

### 3.1. Handcrafted Approaches

As described in Figure 2, handcrafted approaches established the foundation for video-based human action recognition, which entails a manual feature engineering process, where human experts manually design features that support a computer to understand.

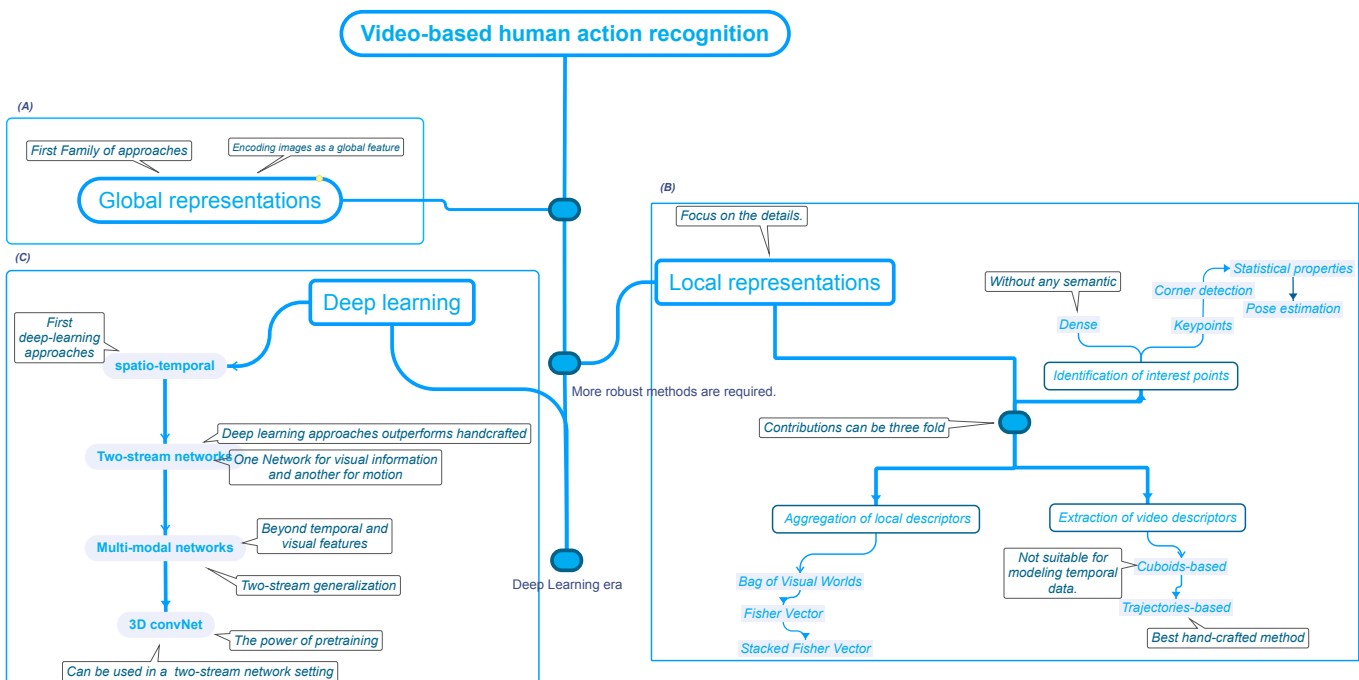

**Figure 2.** The Evolution of Action Recognition Approaches. The initial attempt at vision-based human action recognition relied on global representations (**A**), which were inferior to local representations (**B**). Lastly, deep learning approaches (**C**) became the most popular, with 3D convolutional neural networks becoming the most advanced because they can learn multiple levels of representations.

Two main components usually form handcrafted approaches. Firstly, feature extraction [4] transforms the input video into a video representation of the action. Secondly, the Action Classification [4] component maps the video representation onto a label tag.

### 3.1.1. Feature Extraction

Global representations [20] are the first attempt to recognize actions whose intuition is to capture the video input into one global feature. A simple intuition of the effects of

this type of method is our natural ability to recognize human actions only by looking at the subject's silhouette. However, this approach proved inadequate in addressing the numerous challenges posed by videos or images, such as different viewpoints and occlusions. Consequently, global representations could not fully capture the variability of an action. Among the most relevant methods are Motion Energy Image (MEI) [68], Motion History Image (MHI) [69], silhouettes [70], and Spacetime Volume (STV) [71].

The world is full of little details that are difficult to capture using the "big picture". Intuitively, as humans, to discover those little secrets, we need to explore, focus on the details, and zoom in on the regions of interest, which is the idea behind local representations [21,72], as shown in Figure 2B. Local representations seek to extract descriptors from multiple regions of the video to obtain insights into the details. Local approaches break down into a sequence of steps: (a) detection of points of interest, (b) extraction of video descriptors, and (c) aggregations of local descriptors. As a consequence, the researcher's contributions can be three-fold.

As the name suggests, the first step is to detect which regions of the video to analyze. Nevertheless, determining the significance of a region can be a relatively tricky undertaking. Applying edge detection algorithms is one method, such as Space-Time Interest Points (STIPs) [73] and hessian detector [74]. However, its application could lead to noise and lousy performance due to the extraction of edges that belong to something other than the target subject. To assess the regions' relevance and eliminate noisy information, Liu et al. [75] propose using statistical properties as a pruning method.

Camarena et al. [76,77] suggest that pose estimation can be used as the regions of interest, resulting in a method that has a fixed and low number of processing areas, which ensures a consistent frame processing rate. However, the approach is dependent on the subject body's visibility.

Another solution is to apply dense sampling [78], which consists of placing points without semantics. Dense sampling increases the classification accuracy, but it is computationally expensive [76]. In addition, noise injected by other motion sources can affect the classifier's performance [76,77].

Once we have determined which regions to analyze, we must extract the corresponding region description. Visual and motion data are essential for accurately characterizing an action [76]. In this regard, the typical approach combines several descriptors to have a complete perspective of the target action. Regarding the visual information, we have a Histogram Of Oriented Gradients 3D (HOG3D) [79], Speed-Up Robust Features (SURF) [80], 3D SURF [74], and pixel pattern methods [81–83]. On the other hand, descriptors that focus on motion information include Histogram of Oriented Flow (HOF) [84], Motion Boundaries Histogram (MBH) [78], and MPEG flow [85].

Capturing motion information is a complex task; videos are composed of images in which the target person moves or changes location over time [78]. The naive method uses cuboids, which utilize static neighborhood patterns throughout time. However, cuboids are unsuitable for modeling an object's temporal information. Its natural evolution was trajectory-based approaches [78,86,87] that rapidly became one of the most used methods [21,77].

Trajectory-based methods use optical flow algorithms to determine the position of the object of interest in the next frame, which helps to improve the classification performance [21]. Although several efficient optical flow algorithms exist, their application at different points of interest can be computationally expensive [77]. To reduce the computational time, it is essential to know that there are several motion sources besides the subject of interest, including secondary objects, camera motions, and ambient variables. Focusing on the target motion may reduce the amount of computation required. On the one hand, we can use homographies [21] for reducing the motion's camera; on the other hand, pose estimation [77] can be used to remove the optical flow process thoroughly.

Descriptor aggregation is the final stage in which the video descriptor is constructed using the region descriptors acquired from the preceding processes. There are several

methods, including Bag-of-Visual-Words (BoVW) [88], Fisher Vectors [89], Stacked Fisher Vector (SFV) [90], Vector Quantization (VQ) [91], Vector of Locally Aggregated Descriptors (VLAD) [92], Super Vector Encoding (SVC) [93]. Among the handcrafted approaches, it is popularly referred to as FV and SFV, along with dense trajectories achieving the best classification performance [20].

### 3.1.2. Action Classification

Action classification aims to learn a mapping function to convert a feature vector to a label tag. The literature exposes different approaches, including template-based [68,158,159], generative [160,161], and discriminative models [20,78].

Template-based models are the naive method that compares the feature vector to a set of predefined templates to assign the label tag of the closest instance given a similarity measure. The generative models [160,161] are based on probability and statistics techniques; some representative works include Bayesian Networks and Markov chains.

Discriminative models are one of the most common techniques, including most machine learning methods [20,78]. Due to its performance, handcrafted approaches commonly rely on Support Vector Machines (SVM).

Researchers rely on dimensionality reduction techniques [94] to lower the model's complexity and extract meaningful feature vectors that boost the performance in high-dimensional datasets. Standard techniques include Principal Component Analysis (PCA) [95] and Linear Discriminant Analysis (LDA) [96]. On the one hand, PCA assists in identifying the most representative features, while LDA aids in finding a linear combination of feature vectors that distinguish different action classes.

### 3.2. Deep Learning Approaches

Due to their strong performance in various computer vision tasks [1–3], Convolutional Neural Networks (CNNs) have become increasingly popular. Hence, its application to vision-based human action recognition appeared inevitable.

Andrej et al. [97] developed one of the first approaches, which involved applying a 2D CNN to each frame and then determining the temporal coherence between the frames. However, unlike other computer vision problems, using a CNN does not outperform handcrafted approaches [27]. The main reason was that human actions are defined by spatial and temporal information, and using a standalone CNN does not fully capture the temporal features [27]. Therefore, subsequent deep learning research for human action recognition has focused on combining temporal and spatial features.

As a common practice, biological processes inspire computer vision and machine learning approaches. For example, as individuals, we use different parts of our brain to process the appearance and motion signals we perceive [98,99]. This understanding can be used for human action recognition, as suggested by [98]. The concept is straightforward. On the one hand, a network extracts spatial characteristics from RGB images. On the other hand, a parallel network extracts motion information from the optical flow output [98]. The network can effectively process visual information by combining spatial and temporal information.

Due to the comparable performance of two-stream networks to trajectory-based methods [27], interest in these approaches grows, leading to novel research challenges such as how to merge the output of motion and appearance features. The most straightforward process, referred to as late fusion [100], is a weighted average of the stream's predictions. More sophisticated solutions considered that interactions between streams should occur as soon as possible and proposed the method of early fusion [100].

Because of the temporal nature of videos, researchers investigated the use of Recurrent Neural Networks (RNN) [101] and Long-Term Short-Term Memory (LSTM) [102,103] as the temporal stream for two-stream approaches. As proven by Ma et al. [104], pre-segmented data are necessary to fully explore the performance of an LSTM in videos thoroughly,

eventually leading to Temporal Segment Networks (TSN), which has become a popular configuration for two-stream networks [27].

A generalization of two-stream networks is multi-stream networks [27], which describe actions using additional modalities such as pose estimation [105], object information [106], audio signals [107], text transcriptions [108], and depth information [109].

One factor that impacts the performance of deep neural networks is the amount of data used to train the model. In principle, the more data we have, the higher our network performance. However, the datasets employed in vision-based human action recognition [46,55,110] do not have the scale that requires a deep learning model [48]. Not disposing of enough data has various implications, one of which is that it is difficult to determine which neural network architecture is optimal. Carreiera et al. [48] introduced the Kinetics dataset as the foundation for re-evaluated state-of-the-art architectures and proposed a novel architecture called Two-Stream Inflated 3D ConvNet (I3D) architecture, based on 2D ConvNet inflation. I3D [48] demonstrates that 3D convolutional networks can be pre-trained, which aids in pushing state-of-the-art action recognition further. Deep learning methods work under a supervised methodology implicating considerable high-quality labels [111]. Nevertheless, data notation is a time-intensive and costly process [111]. Pretraining is a frequent technique to reduce the required processing time and amount of labeled data [111]. Consequently, researchers explored the concept of 2D CNN inflation further [112,113], yielding innovative architectures such as R(2+1)D [114].

Current research in vision-based human action recognition has several directions. First, novel architectures such as visual transformers have been ported to action recognition [114,115]. Second, there is a need for novel training methods such as Self-Supervised Learning (SSL) [43], which is a novel training technique that generates a supervisory signal from unlabeled data, thus eliminating the need for human-annotated labels. Third, few-shot learning action recognition is also being investigated [44].

Most of the architectures described are known as discriminative approaches [116], but there is another family of deep learning methods based on generative techniques [116]. Its core idea is based on the popular phrase "if I cannot create it, then I do not understand it" [117]. Auto-encoders [118], variational autoencoders [119], and Adversarial Networks (GAN) [120] are examples of this approach.

## 4. Current Research and Future Directions

The video-based human action recognition field is currently undergoing promising research in multiple directions that will shape its future directions.

### 4.1. New Video-Based Architectures

Since the growing popularity of transformers in natural language processing [25] due to their outstanding capability to process sequential data and superior performance to the well-known convolutional neural networks (CNN) in image-related tasks [25], researchers have been exploring the benefits of visual transformers for human action recognition in video-based applications.

Human action is defined in a visual and temporal space, and understanding the sequential information became crucial to comprehend individual actions and the relationships and dependencies between them [71]. Nevertheless, current methods only focus on the short time frame, which limits the understanding of the impact and consequences of action in the long term, a crucial aspect for the model deployment in real open-world scenarios [22]. Hence, novel architectures should improve an action's visual and temporal information, improving the classification performance.

### 4.2. Learning Paradigms

A second direction relates to the learning paradigms used to train a model, where supervised learning [42] is the most common; a supervised methodology requires a labeled dataset, meaning every action sample passes through a human-annotated process [50].

Unfortunately, this labeling process is costly and time-consuming, particularly in high-dimensional datasets needed for deep learning approaches [50].

Despite the performance of supervised learning, researchers started to explore new approaches, including semi-supervised learning [121], weakly-supervised learning [122], and Self-Supervised Learning (SSL) [42,43].

Weakly-supervised learning [122] leverages the related information and metadata available on social media, such as hashtags, to approximate the action label tag. On the other hand, the core idea of semi-supervised learning [121] is to extract visual features relying on a small-scale labeled dataset and a large-scale unlabeled dataset. Finally, Self-Supervised Learning (SSL) [42,43] extracts the supervisory signal using the unlabeled data based on the intuition of a child's capability to learn by exploring and interacting with the world.

There are two prominent families of SSL approaches: pretext tasks and contrastive learning. Pretext tasks [123] involve defining an auxiliary function that provides supervised signals without human annotation. For example, a network may be asked if a sequence of video frames is in the correct order. On the other hand, contrastive learning for SSL [162] aims to identify differences between video samples by projecting them onto a shared feature space where clips from the same distribution are clustered together based on a distance metric. Pretext tasks and contrastive learning can be used together, as Pretext Contrastive Learning (PCL) suggests [123]. PCL combines a pretext task function to capture local information and contrastive learning loss functions to gain a global view.

Another challenge related to training a deep learning model is the high correlation between the data used in training and the model performance [50]. On the one hand, it is impractical to construct a novel dataset for each required task. Second, some application domains require highly specific data, such as medical records, making it difficult to recollect a high amount of data [45]. Therefore, few-shot action recognition aims to create models that generalize efficiently in low-data regimes [44].

Its motivation is threefold [124–126]: to enable learning representations for applications where acquiring even unlabeled data is complex, to reduce the high computational demand required for processing large datasets, and to generalize novel action classes not presented in the training dataset. While most few-shot learning research has focused on image tasks [127–129], its extension to video classification is still an open question and remains largely unexplored [125,130].

*4.3. Pretraining and Knowledge Transfer*

Transferring knowledge from one model to another is a standard technique to reduce computational resources and dependency on labeled data [42]. Traditional methods include transfer learning [131] and fine-tuning [132], which leverage the multi-level representations from deep learning architectures. Transfer learning [131] starts with pre-trained network weights, while fine-tuning [132] adds trainable layers to an existing model. Nevertheless, both transfer methods are model-agnostics meaning that the transfer depends on the model architecture and objective tasks [50]. Novel methods have been explored, including knowledge distillation [50,51,133] that uses a teacher–student framework that enables the transfer even when the new network does not share the same architectural design.

Additionally, to transfer knowledge between different architectural designs, researchers suggest that new methods can lead to transfer learning between different input modalities [134–136]. Sharing knowledge between modalities is challenging, and explorations using disjunct but natural modalities, such as text and visual information, remain a future direction [134]. In addition to using different modalities, novel directions focus on constructing visual models that enable the extraction of visual features that can be representative across multiple video domains in addition to action recognition to construct unified pretraining models [137].

### 4.4. Video Modalities

Most of the works discussed employ RBG modality; however, incorporating other modalities can benefit various applications scenarios for video-based human action recognition [25]. Modalities can be divided into visual and non-visual [25].

Among visual modalities are RGB [21,134], Skelethon [138,139], depth [24], infrared [140], and thermal [141], each with strengths. For example, the depth [24] modality can extract the objects' shape and structure from the scene. Conversely, infrared [140] modality can capture information in low-light or no-light conditions, and thermal [141] information can detect hidden objects such as humans and temperature monitoring.

One of the most used modalities in video-based human action recognition is skeleton data [138,139,142], which aims to understand human actions using the sequence of the subject skeleton. In contrast to traditional RGB, where Convolutional Neural Networks (CNN) are the standard technique [143], skeleton-based action recognition relies on Graph Neural Networks (GCN) [144,145]. Ref. [143] compares convolutional neural networks to Graph Neural Networks (GCN), showing that proper training techniques, augmentations, and optimizers lead to comparable performance. Ref. [139] presented PYSKL: an open-source toolbox for skeleton-based action recognition that, in addition, to providing CNN and GCN implementations, established a set of good practices to ease the comparison of efficacy and efficiency. Ref. [145] retakes the idea of multiple stream networks and proposes the GCN-Transformer Network (ConGT), which extracts spatial information using the Spatial-Temporal Graph Convolution stream (STG) and temporal information using the Spatial-Temporal Transformer stream (STT).

Regarding non-visual modalities, there are several options, such as audio [146], acceleration [147], radar [148], and WiFi [149], and they are mainly used as complementary data and privacy enhancement. For instance, audio [146] is widely captured along with visual data by video cameras, and it can provide additional and more representative information about some actions, including detecting anomaly events. Radar [148] and WiFi [149] signals 3D-map the environment and understand the object's motion and position in the scene, even in ambient conditions with high levels of occlusions. Finally, the acceleration [147] modality leverages our daily devices' sensors to extract information about motion and body orientation.

In addition to the modalities presented, some may complement our understanding of human action. For example, despite the significant variability of our actions, they have physical limitations, both human and environmental [22]. For this reason, codifying physical properties could lead to a greater understanding of human actions.

### 4.5. Multi-Modal and Cross-Modal Learning

Speaking about video modalities, our interaction with the world is multi-modal [13], meaning we interact using multiple sensorial inputs.

Therefore, there is no reason that current models use a unique modality. Consequently, leveraging multiple modalities became a new research direction to use the strength of use modality to improve the performance and robustness.

There are two main approaches for using multi-modal learning: multi-modal [43,150] and cross-modal [151]. The foundation of multi-modal learning [43,150] is that diverse modalities can extract different and complementary information that, in conjunction, results in a complete comprehension of the action sample [24]. There are two primary types of approaches to multi-modal learning: fusion [152–154] and co-learning [154]. Fusion [152,153] methods involve merging the classification outputs of models trained separately in different modalities, which can be challenging. In contrast, co-learning [154] aims to use modalities in conjunction with training instead of using them independently, which is more natural to our world perception.

On the other hand, not all modalities are always available simultaneously or are as easy to extract as others [134]. Therefore, cross-modal action recognition aims to transfer knowledge from models trained on different modalities [134], leading to some advantages,

including boosting the performance of a uni-modal model or weaker modality using a stronger modality. Additionally, cross-modal may improve the performance in low-data scenarios [134].

### 4.6. Explainability

Although deep learning models lead the state-of-the-art in video-based human action recognition, the model outputs are often considered "black boxes" [155], meaning that it is difficult to understand how they make decisions. Some application domains, including video surveillance, imply decisions have real-work consequences; being able to explain the model output is required to be trusted by humans and build transparency in any ethical concern [22]. Explainability in video-based human action recognition is challenging in addition to visual information, and it should include the ability to explain temporal timeframes [22].

It is essential to mention that the current research directions are not standalone paths, and research that combines them is relevant. For example, self-supervised learning is complementary to few-shot learning [156]. Furthermore, knowledge distillation can be used for cross-modal transfer learning [134]. Other relevant directions include studying new data augmentation techniques [157], neural architecture search [4], and efficient network training methods [26]. In addition, constructing a novel dataset that supports previous research directions remains critical to developing novel methods [27].

## 5. Conclusions

This work provides an overview of the video-based human action recognition field. We started by defining core concepts of the field, including the definition of what an action is and the goal of video-based human action recognition. Then, we described the challenges of action-class variability, sensor capabilities, environment conditions, dataset restrictions, and computational constraints explaining their implications and possible consequences. Finally, we introduced some of the most used datasets in the literature, including traditional RGB-based datasets such as KTH, Weizmann, UCF101, HDMB51, Kinetics, ActivityNet, and YouTube-8M. In addition, we found some datasets that provide additional modalities inputs such as NTU RGB+D, Kinetics-Skeleton, and J-HMDB. The information presented may help future works to have a shared understanding of the fundamental ideas of the field.

Conversely, to provide researchers with a clear intuition of what has been explored and make complex concepts accessible, we explore the approaches proposed in the literature and break down their evolution over time, emphasizing the intuitive notions that underpin the approaches' advancements. The explorations include traditional handcrafted and deep learning approaches. We described local and global feature extraction methods and some standard action classification techniques regarding handcrafted methods. Regarding deep learning methods, we explored traditional methods, two-stream networks, and 3D CNN.

Finally, we explored current research directions and potential paths to help future works identify novel opportunities and boost the process of constructing meaningful contributions. We divided the directions into six blocks: implementation of new architectures, new learning paradigms, new pretraining and transfer methods, exploration of novel modalities, multi-modal and cross-modal, and finally, the model explainability.

**Author Contributions:** Conceptualization, L.C.; formal analysis, F.C.; investigation, F.C.; methodology, F.C. and M.G.-M.; project administration, M.G.-M.; resources, M.G.-M.; supervision, M.G.-M. and L.C.; validation, M.G.-M.; writing—original draft, F.C. and R.C.-A.; writing—review and editing, F.C. and R.C.-A. All authors have read and agreed to the published version of the manuscript.

**Funding:** F.C. gratefully acknowledges the scholarship no. 815917 from CONACyT to pursue his postgraduate studies. The scholarship had no role in the study design, data collection and analysis, decision to publish, or preparation of the manuscript. The authors would like to thanks the financial support from Tecnologico de Monterrey through the "Challenge-Based Research Funding Program 2022". Project ID E120-EIC-GI06-B-T3-D.

**Conflicts of Interest:** The author declares that he has no conflict of interest.

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
