# Peer review of "An Overview of the Vision-Based Human Action Recognition Field"

_mca, doi:10.3390/mca28020061_

Round 1
Reviewer 1 Report
- The contribution of this document is to present the information in a comprehensive and easy to understand way, work and evolution of the area of ​​recognition of human actions.
-I suggest that the font in Figure 1 be larger and to be more understandable.
- Reference is made first to Figure 2 and then to Figure 1. The order of them must be corrected.
-I would question the depth with which the authors comment that the paper is presented, because in my opinion, there is only one figure that tries to illustrate it, and being very large, it is difficult to understand.
-When authors talking about depth on the subject, I think authors could talk about algorithms, pseudocodes, tools that can help in the recognition of human actions.
- I question whether the authors use the keywords most commonly used in research papers in recent years in the area of ​​recognition of actions, since the words that appear the most are "human", "action, "video", "recognition", and I think not provide relevant information to identify trends and predict possible future directions as stated by the authors. Instead, I would suggest conducting a Systematic Review of the Literature guided by research questions.
-In the abstract, the authors mention that "this concise survey helps researchers understand the breadth of the approach..." but throughout the papers it is mentioned that its goal is an in-depth presentation of the topic.
-In general, the subject is of interest and not trivial, however, I believe that in order to provide an overview that allows researchers to make decisions, a more in-depth review should be done. For example, presenting research results by papers, expert authors in the topic, since the reviewed articles are not even presented. It would be good to indicate what type of CNN they use, if compared to other methods, or in the case of optical flow, they use Lucas & Kanade's or a dense one like Horn and Schunk's. This type of depth would determine the lines of investigation to follow and make decisions.
Author Response
The attached PDF contains our response

Reviewer 2 Report
The authors provide a concise exploration of the video-based human action recognition field, listing the available techniques, their evolution throughout the years and possible future directions. The paper is well writen, with a few minor errors. Some points deserve special attention in order to improve paper quality and readers' understanding.
Despite the fact the the research performed by the authors is interesting, I believe it could be more extensive. For instance, the authors should have included the other webservices to extract research data, as the ones mentioned (arXiv, scopus, science direct, etc.). Also, more terms should be used in the query, for instance, "human object interaction". The authors should clearly present the inclusion and exclusion conditions. Did the authors considered papers from different languages or only in English? This is not clear in the text. Also, was the paper length taken into consideration? This should be explained in more details, so that a reader may have more information if he/she desires to reproduce the work and extend it to the new years to come.
Important references are missing and should be cited in the text:
- Pyskl: Towards good practices for skeleton action recognition
- Spatial Temporal Graph Convolutional Networks for Skeleton-Based Action Recognition
- MotionBERT: Unified Pretraining for Human Motion Analysis
- Channel-wise Topology Refinement Graph Convolution for Skeleton-Based Action Recognition
It seems that a evaluation based only in the papers metadata is not enough. I would expect a deeper evaluation comparing the main works in the state of the art, recommending the algorithms according to the range and characteristics of the activities that should be recognized.
Some general comments and minor errors are listed as follows.
"This survey provides an in-depth exploration" -> I believe, according to the title, this is a very concise overview of the field
"methodologies. While [18]" -> "methodologies, while [18]"
"Deep learning approaches outperforms" -> "Deep learning approaches outperform"
"and some keypointsld." -> ?
" knowns as" -> " known as"
"use of statistical" -> "use statistical"
"object. its " -> "object. Its "
"that rapidly becoming" -> "that rapidly became"
" trajectories, achieve" -> " trajectories, achieving"
"methods, [20]," -> "methods [20],"
" is necessary to explode" -> "explode" or "explore"?
"Do not dispose" -> "Do not disposing"
"Pretrained is a frequent technique" -> "Pretraining is a frequent technique"
"auto-encoders [75]," -> "Auto-encoders [75],"
"APIs" -> the term should be defined only once in the text
"easy-to-understand manner," -> "in an easy-to-understand manner,"
"high-quality labels. Assuming" -> "high-quality labels, assuming"
Author Response
The PDF contains our respond to the reviewers comments

Round 2
Reviewer 1 Report
I confirm that the authors responded to the suggestions
Reviewer 2 Report
I'm satisfied with the modifications performed by the authors. Congratulations, you did a great job and I believe the paper is ready for acceptance. As this is a survey paper, please modify the submission type accordingly, as it is classified as Article instead of Survey.